# Immunogenicity and Protective Efficacy of a Recombinant *Toxoplasma gondii* GRA12 Vaccine in Domestic Cats

**DOI:** 10.3390/vaccines13080851

**Published:** 2025-08-11

**Authors:** Jinru Yang, Linchong Nie, Yining Song, Zipeng Yang, Liulu Yang, Hongjie Ren, Wenhao Li, Yasser Mahmmod, Xiu-Xiang Zhang, Ziguo Yuan, Hao Yuan, Yan Zhang

**Affiliations:** 1College of Science, Shenyang University, Shenyang 110044, China; 17770052806@163.com; 2Key Laboratory of Zoonosis Prevention and Control of Guangdong Province, College of Veterinary Medicine, South China Agricultural University, Guangzhou 510642, Chinaziguoyuan@scau.edu.cn (Z.Y.); 3College of Agriculture, South China Agricultural University, Guangzhou 510642, China; xiuxiangzh@scau.edu.cn; 4Department of Veterinary Clinical Sciences, Lewyt College of Veterinary Medicine, Long Island University, 720 Northern Boulevard, Brookville, NY 11548, USA; yasser.mahmmod@liu.edu

**Keywords:** *Toxoplasma gondii*, rGRA12, immune protection effect, cat

## Abstract

**Background:** *Toxoplasma gondii* (*T. gondii*) is a significant opportunistic zoonotic protozoan, presenting a substantial risk to human health and livestock. Consequently, the development of an effective vaccine against toxoplasmosis is imperative. This study focuses on the GRA12 protein as a target for developing a recombinant protein vaccine, with its efficacy evaluated through immunization trials in cats. **Methods:** We expressed recombinant GRA12 protein in *E. coli* and immunized cats with the purified antigen. The cats were categorized into four groups: G1 (PBS control), G2 (ISA 201 adjuvant alone), G3 (rGRA12 vaccine), and G4 (rGRA12 combined with ISA 201 adjuvant). All cats underwent subcutaneous immunizations on days 0, 14, and 28. Subsequently, serum levels of IgG (including IgG1 and IgG2a subclasses) and cytokines (IFN-γ, IL-2, TNF-α, IL-4, IL-10) were measured by enzyme-linked immunosorbent assay (ELISA). Two weeks after the third immunization (42 DPI), each cat was intraperitoneally infected with 1 × 10^6^
*T. gondii* RH tachyzoites. Oocyst shedding, survival duration, and *T. gondii* burden were monitored to assess vaccine-induced immunity. **Results:** The results indicate that immunization with recombinant rGRA12 protein significantly elevated IgG, IgG1, and IgG2a antibody levels in cats. G4 displayed elevated IgG levels post-immunization compared to G1 and G2, with an IgG1/IgG2a ratio > 1, indicating a mixed Th1/Th2 immune response. G4 also showed significantly increased IFN-γ, IL-2, TNF-α, and IL-4 levels compared to G1 (*p* < 0.05), while IL-10 remained unchanged. After *T. gondii* infection, total oocyst counts were 4.61 × 10^6^ (G1), 4.49 × 10^6^ (G2), 3.58 × 10^6^ (G3), and 2.59 × 10^6^ (G4), with G3/G4 showing 20.1–27.9% reduction relative to G1 (*p* < 0.05). Survival analysis revealed that groups G3 and G4 exhibited significantly longer median survival times (38 and 60 days, respectively; G4 with no mortality) compared to G1 and G2 (19 and 26 days, respectively). Additionally, parasite burdens in the brain, heart, lungs, liver, and spleen were significantly reduced in G3/G4 compared to G1/G2 (*p* < 0.01). **Conclusions:** In summary, the recombinant GRA12 vaccine significantly enhanced host survival and reduced parasite burden, demonstrating its potential as an effective toxoplasmosis vaccine candidate. These findings provide valuable data for future toxoplasmosis vaccine development.

## 1. Introduction

*T. gondii*, an obligatory intracellular protozoan, exhibits a pronounced tropism for nucleated cells in virtually all endothermic vertebrate taxa, including human [1]. Humans and animals are commonly infected by eating undercooked meat containing tissue cysts or by consuming food or water contaminated with sporulated oocysts from feline feces [2]. While infections often remain asymptomatic in individuals with healthy immune systems, immunocompromised people are at much higher risk for serious conditions like encephalitis, myocarditis, and pneumonia [3,4]. If a pregnant woman acquires an acute infection, the parasite can cross the placenta and result in serious pregnancy outcomes, such as miscarriage, intrauterine fetal death, premature birth, or congenital defects [5]. In livestock, *T. gondii* also causes major reproductive losses, particularly abortions in economically significant species like swine and sheep [6,7].

Felids, particularly domestic cats, serve as the definitive hosts for *T. gondii* and are the only animals capable of shedding environmentally resistant oocysts. A single cat can release up to 55 million oocysts per day during the acute stage of infection [8,9]. Global economic growth and urbanization have contributed to rising domestic cat populations, amplifying zoonotic transmission risks. Meta-analysis of studies from 1967 to 2017 revealed a global *T. gondii* seroprevalence of 35% in domestic cats and 59% in wild felids [10]. In certain regions of China, such as Jilin and Shandong, pet owners had *T. gondii* infection rates of 18.10% and 18.04%, respectively, significantly higher than the general population’s rate of 7.88% [11,12]. Similarly, a serological study in western Thailand found that dairy cows exposed to pets were four times more likely to test positive for *T. gondii* than cows without such exposure [13,14]. These findings highlight the urgent need to develop safe and effective vaccines targeting feline hosts to help control the spread of toxoplasmosis.

*T. gondii* contains three specialized apical secretory organelles—micronemes, rhoptries, and dense granules—that play key roles during host cell invasion. As the parasite’s tachyzoites form invades, microneme proteins (MICs) are secreted from the apical end to facilitate recognition and attachment to the host cell membrane. Rhoptry proteins (ROPs) then act in coordination with MICs to assist in host cell entry and the formation of a protective parasitophorous vacuole (PV). Following this, dense granule proteins (GRAs) are secreted to modify the PV environment, allowing for nutrient acquisition that supports parasite replication [15].

Among the GRAs, GRA12 behaves similarly to GRA2 and GRA6 by being released into the PV shortly after invasion. It first localizes to the posterior invaginated pocket to assemble the nanotubular network, eventually spreading throughout the vacuolar space, where it is present in both soluble and membrane-bound forms [16]. GRA12 has 53 predicted post-translational modification sites, a transmembrane domain, and multiple predicted B-cell and T-cell epitopes. Immunological analyses suggest that GRA12 is strongly immunogenic and non-allergenic, making it a promising vaccine target [17]. Notably, deletion of the GRA12 gene severely reduces chronic-stage cyst development in vivo, although it does not affect parasite growth or cyst differentiation in vitro [18]. Overall, this study aims to demonstrates the immunogenicity and protective potential of the recombinant GRA12 (rGRA12) subunit vaccine, including its capacity to reduce *T.gondii* transmission in felids and mitigate host damage, providing a scientific basis for further Toxoplasma vaccine development.

## 2. Materials and Methods

### 2.1. Ethics Statement

The animal experiments were approved by the Laboratory Animal Ethics Committee of South China Agricultural University (Ethics Number: 2024F069), and the experimental animals were raised at the Laboratory Animal Center of South China Agricultural University.

### 2.2. Cats

To maximize animal welfare while maintaining scientific validity, we implemented the 3Rs principle (Replacement, Reduction, and Refinement) throughout our experimental design. This included using the minimum number of animals required for statistically significant results, optimizing protocols to minimize discomfort. All the care procedures were approved by the Animal Ethics Committee of South China Agricultural University (no.SCAU2022f208). Twelve 4-month-old British Shorthair cats (2–3 kg) were enrolled in this study. The animals were previously monitored for 2 months prior to the beginning of the experiment. The cats were randomly allocated to individual cages and provided ad libitum access to commercially available cat food and fresh drinking water. Serum was collected from all cats one week pre-experiment to verify naive status. Modified agglutination testing (MAT) and sucrose flotation techniques were employed to rule out prior *T. gondii* infection.

### 2.3. Parasite Propagation and Harvest

Tachyzoites of the *T. gondii* RH strain were obtained from the Laboratory of Parasitology at South China Agricultural University. These tachyzoites were used to infect human foreskin fibroblast (HFF) cells maintained in medium supplemented with 2% serum. Once 80% of cells exhibited cytopathic effects, they were detached using a cell scraper. The resulting suspension was homogenized by passing it 20–25 times through a 27-gauge needle. The homogenate was then subjected to two centrifugation steps: first at 1500× *g* for 5 min at 4 °C to remove cellular debris, and then at 3000× *g* for 7 min at 4 °C to collect the tachyzoites. The final pellet was resuspended in sterile phosphate-buffered saline (PBS), and concentration of tachyzoites was quantified using hemocytometry [19].

### 2.4. Construction of Plasmids, and Expression and Purification of rGRA12

The protocol for obtaining rGRA12 was conducted following the method described by Wang [20]. The DNA sequence of the *T. gondii* GRA12 gene was obtained from the GenBank database (accession number: FJ011096.1). Genomic DNA was isolated from *T. gondii* RH strain tachyzoites and used as a template to amplify the GRA12 gene via standard PCR. Based on its sequence, the predicted molecular weight of GRA12 antigen was 47.8 kDa. The 1311 bp GRA12 fragment was amplified using primers GRA12F (5′-TGAGCTCATCATGAGGGCGATCGTGGCATCGACG-3′, *SacI* site underlined) and GRA12R (5′-CAAGCTTGTTGTGTTTGCTGCCTGCAGAGCCGCG-3′, *HindIII* site underlined), and then cloned into the pET-28a vector via *SacI*/*HindIII* restriction sites. The recombinant plasmids were verified by *SacI*/*HindIII* double digestion and sequencing of the gel-purified fragments. Sequence-confirmed positive clones were subsequently transformed into *E. coli* BL21 for protein expression. *E. coli* BL21(DE3) harboring pET-28a-GRA12 was inoculated from a 5 mL starter culture into 500 mL of kanamycin-supplemented LB medium. The culture was grown at 37 °C with 170 rpm shaking to OD_600_ 0.4–0.5, at which point protein expression was induced with 0.1% IPTG. Following 4 h of continued shaking under identical conditions, the bacterial cells were harvested for subsequent protein purification. The purified recombinant GRA12 protein (47.8 kDa) was subsequently used for feline immunization.

### 2.5. Grouping, Immunization, and Challenge Infection of Cats

The GRA12 protein was heterologously expressed and purified in our laboratory. A total of twelve 6-month-old female domestic cats with blue coats were randomly divided into four groups (three cats per group): G1 (PBS control), G2 (ISA 201 adjuvant alone), G3 (rGRA12 vaccine), and G4 (rGRA12 combined with ISA 201 adjuvant). Subcutaneous immunizations were administered three times at two-week intervals. Table 1 and Figure 1 provides a clear overview of the immunization timeline. Blood samples collected on days 0, 14, 28, and 42, and serum was isolated for the evaluation of antibody titers and cytokine concentrations.

All cats were challenged on day 42 with 1 × 10^6^ tachyzoites of the *T. gondii* RH strain. Survival was monitored daily for 60 days, with group-specific survival times recorded.

### 2.6. Enzyme-Linked Immunosorbent Assays for IgG, IgG1, and IgG2a

Serum samples were collected from all groups on days 0, 14, 28, and 42 following immunization. The rGRA12 antigen was prepared at a concentration of 10 μg/mL in a carbonate–bicarbonate coating buffer (50 mM, pH 9.6). A 96-well plate was coated with 100 μL of this antigen solution and incubated overnight at 4 °C. The plate was then washed three times with PBST (PBS containing 0.05% Tween-20), and nonspecific sites were blocked using 100 μL of 1% BSA per well at 37 °C for 1 h. Serum samples were then diluted 1:400 in PBS, and 100 μL of each diluted sample was added to the wells. After incubation at 37 °C for 1 h and another three washes with PBST, 100 μL of horseradish peroxidase (HRP)-conjugated secondary antibody (diluted 1:2000 in PBS) was added and incubated for an additional hour at 37 °C. The plate was then washed five times with PBST, and 100 μL of TMB substrate was added to initiate color development. After a 10 min incubation in the dark, the reaction was stopped by adding 50 μL of 2 M sulfuric acid. Absorbance at 450 nm was measured within 15 min using a microplate reader. All tests were performed in triplicate. Total IgG levels were measured on days 0, 14, 28, and 42, while IgG1 and IgG2a subclass levels were specifically assessed on day 42.

### 2.7. Cytokine Assays

To detect cytokines on day 42 post-immunization, blood was collected from vein. The levels of IL-2, IL-4, IL-10, IFN-γ, and TNF-α were examined using commercial ELISA kits, according to the manufacturer’s instructions (Shanghai Meilian Biotechnology Co., Ltd., Shanghai, China). Data from three independent experiments were analyzed, including three replicates per serum. All procedures were performed according to the manufacturer’s detailed protocols.

### 2.8. Quantitative Detection Process of Cat Oocysts

Fresh fecal samples were homogenized in distilled water (1:10) within 24 h of collection. One gram of the homogenized material was mixed with 10 mL of saturated sucrose solution, vortexed for 30 s, and filtered through a 100 μm mesh. The filtrate was centrifuged at 1200× *g* for 10 min at 4 °C. The upper layer of the sucrose solution was gently aspirated using a Pasteur pipette. For samples that tested positive for oocysts under microscopy, a secondary purification step was conducted: 9 mL of the supernatant was diluted with 40 mL of deionized water and centrifuged again under the same conditions. The supernatant was discarded, and the pellet was resuspended in 1 mL of distilled water. The final suspension was loaded onto a modified McMaster slide, allowed to settle for 3 min, and examined under a light microscope to count the oocysts. Oocyst counts were used to calculate the number of oocysts per gram of feces, which served as an estimate of total oocyst excretion. Group-wise daily average oocyst shedding was statistically analyzed, and kinetic shedding curves were constructed. Protective Fraction (PF) was calculated using the following formula: PF = (P2 − P1)/P2, where P2 was the mean count in the control group and P1 was the mean count in the immunized group.

### 2.9. Quantitative Analysis of Parasitic Burden in Tissues

Experimental procedures were performed as follows: Primers targeted the *T. gondii* B1 gene (F: 5′-TCCTTCGTCCGTCGTAAT-3′; R: 5′-TTCTTCAGCCGTCTTGTG-3) [21]. Genomic DNA extracted from harvested *T. gondii* tachyzoites was used as a template to amplify the B1 gene by PCR. The purified target DNA fragment was ligated into the pMD18-T vector to construct the recombinant plasmid, which was then transformed into DH5α competent cells. Following sequence verification, positive clones were selected for overnight culture in LB medium supplemented with 100 μg/mL ampicillin (37 °C, 220 rpm). DNA concentration and purity were determined by UV spectrophotometry, with A260/A280 ratios between 1.8 and 2.0 considered acceptable. The plasmid was serially diluted to generate RT-qPCR standards (10^9^ to 10^4^ copies/μL). For RT-qPCR, 20 μL reactions comprised 10 μL of 2× SYBR Green qPCR Mix, 0.5 μL of each primer (10 μM), 1.0 μL of template DNA, and 8 μL of RNase free water. Thermal cycling conditions were as follows: initial denaturation at 95 °C for 10 min, followed by 40 cycles of 95 °C for 10 s, 60 °C for 31 s, and 72 °C for 30 s. To quantify tissue parasite burden, brain, heart, lung, liver, and spleen tissues were collected immediately after euthanasia for pathological and molecular analyses.

### 2.10. Statistical Analysis

Statistical analyses were conducted using IBM SPSS Statistics 26.0 with a hierarchical approach (International Business Machines Corporation, New York City, NY, USA): one-way ANOVA followed by Tukey’s post hoc test was applied for multi-group comparisons, while unpaired Student’s *t*-tests evaluated differences between two groups. Statistical significance was defined as * *p* < 0.05, ** *p* < 0.01, and *** *p* < 0.001.

## 3. Results

### 3.1. IgG Antibody Detection

Serum IgG concentrations in four feline cohorts (G1–G4) were quantified by ELISA pre- and post-immunization (Figure 2). Total IgG titers increased progressively after multiple vaccinations, reaching peak concentrations at day 42. G3 and G4 exhibited significantly higher IgG levels than G1 (*p* < 0.01), beginning on day 14 post-immunization, with G4 maintaining significantly higher levels than G3 at days 28 and 42 (*p* < 0.01). However, no statistically significant difference was observed between the G1 and G2 groups (*p* > 0.05).

### 3.2. IgG Antibody Isotype Determination

Serum IgG isotype profiles (IgG1/IgG2a) were characterized to explore potential Th1/Th2 immune polarization. As shown in Figure 3, G1 and G2 showed no statistically significant difference in IgG1 and IgG2a levels at 42 days post-immunization (*p* > 0.05). Vaccination with rGRA12 protein (G3/G4) demonstrated concurrent upregulation of both IgG subclasses (*p* < 0.05), displaying IgG2a predominance over IgG1 (ratio > 1). This isotypic polarization suggests coordinated Th1/Th2 immune activation following antigenic challenge.

### 3.3. Cytokine Production Assay

Serum concentrations of Th1-type cytokines (IFN-γ, IL-2, TNF-α) and Th2-type cytokines (IL-4, IL-10) were quantified following rGRA12 protein administration (Figure 4). By day 42 post-immunization, both G3 and G4 demonstrated significantly elevated IFN-γ, IL-2, and TNF-α levels compared to G1 and G2 (*p* < 0.01) (Figure 4A–C). After immunization with rGRA12 protein, IL-4 levels in G3 remained stable (*p* > 0.05), whereas G4 exhibited a pronounced increase (*p* < 0.01) (Figure 4D). No significant differences in IL-10 concentrations were observed in either G3 or G4 (*p* > 0.05) (Figure 4E). Compared to G3, G4 displayed statistically significant upregulation of IFN-γ, IL-2, TNF-α, and IL-4, while IL-10 levels remained comparable between the two groups (*p* > 0.05).

### 3.4. Feline Oocyst Volume Statistics

On day 42 post-vaccination, cats were intraperitoneally challenged with 1 × 10^6^ RH *T. gondii* tachyzoites. Fecal samples were collected daily from day 1 to day 20 post-infection for oocyst shedding quantification. All infected cats excreted *T. gondii* oocysts; however, total oocyst shedding in G3 and G4 was significantly reduced compared to G1 and G2 (Figure 5B). Shedding dynamic analysis showed distinct patterns: G1 initiated oocyst shedding on day 3 post-infection, persisting for 12 days with a total output of 4.61 × 10^6^ oocysts; G2 commenced shedding on day 3, producing 4.49 × 10^6^ oocysts over 12 days; G3 delayed shedding onset to day 4, with 3.58 × 10^6^ oocysts shed during a 10-day period; G4 exhibited the most delayed onset (day 4) and shortest shedding duration (9 days), yielding 2.59 × 10^6^ oocysts (Figure 5A). Compared to G1, total oocyst shedding in G2, G3, and G4 decreased by 2.62%, 20.10%, and 27.88%, respectively. rGRA12 vaccination significantly suppressed oocyst excretion (*p* < 0.05) and reduced potential environmental transmission risks. Detailed statistical summaries of oocyst shedding parameters are provided in Table 2.

### 3.5. Protective Efficacy of Recombinant Protein Vaccination in Cats

Immunization efficacy was evaluated by survival analysis in *T. gondii*-infected cats. At 42 days post-immunization, all groups were challenged with RH. As shown in Figure 6, the vaccinated groups (G3 and G4) demonstrated significantly higher survival rates compared to both G1 and G2 groups. Importantly, within the vaccinated groups, G4 exhibited superior survival compared to G3. Median survival times were 20 days for G1, 30 days for G2, and 38 days for G3, while G4 achieved 100% survival, with no mortality observed. rGRA12 vaccination, when formulated with ISA 201 adjuvant, enhanced protection against *T. gondii* infection.

### 3.6. Quantification of Parasite Burden in Various Organs of Cats

Absolute RT-qPCR was used to quantify parasite loads in multiple organs (brain, heart, lung, liver, spleen) across all groups. Compared to G1 and G2, cats immunized with rGRA12 alone (G3) displayed significantly reduced *T. gondii* loads across all tested organs (*p* < 0.01) (Figure 7). Notably, G4 demonstrated enhanced protection compared to G3, with statistically lower parasite burdens in every organ analyzed (*p* < 0.05).

## 4. Discussion

In China, the reported 15.3% seroprevalence of *T. gondii* in edible livestock, with regional and temporal variations, highlights the widespread environmental contamination by feline-derived oocysts and its potential impact on food safety [22]. Given the significant health threats posed by *T. gondii* to immunocompromised populations and its considerable economic impact on livestock industries, there is an urgent demand for effective vaccines against toxoplasmosis [23]. Among the existing vaccine strategies, recombinant protein-based vaccines have emerged as safe and potent alternatives to traditional vaccine approaches. These subunit vaccines serve as leading candidates for preventing both acute and chronic *T. gondii* infections, and show great promise for protective use [24].

Research shows that anti-*T. gondii* IgG plays a vital role in preventing parasite adhesion to host cells and works alongside macrophages to clear the parasite [25]. In this study, cats immunized with rGRA12 exhibited a gradual increase in *T. gondii*-specific IgG levels following each immunization, reaching their highest point following the third dose. The vaccinated groups (G3 and G4) had significantly greater IgG titers than the control groups (G1 and G2) (*p* < 0.05). Among the vaccinated groups, G4 demonstrated significantly higher anti-*T. gondii* IgG levels than G3, indicating that the ISA 201 adjuvant enhanced the immunogenicity of rGRA12. Subclass analysis revealed significant increases in both IgG1 and IgG2a in G4, with IgG2a levels surpassing those of IgG1. This pattern suggests that rGRA12 immunization induced a mixed Th1/Th2 response in cats, with a Th1-predominant bias. Previous studies have shown that IgG2a supports Th1-driven adaptive immunity by activating T helper cell responses, whereas IgG1 promotes the differentiation of naive Th cells toward a Th2 phenotype [26].

After rGRA12 immunization, cats exhibited elevated serum concentrations of the pro-inflammatory cytokines IFN-γ and IL-2. IFN-γ, primarily produced by NK cells, CD4^+^ T cells, and CD8^+^ T cells, plays a central role in defending against *T. gondii* by inducing antimicrobial mechanisms such as nitric oxide (NO) and reactive oxygen species (ROS), as well as altering host cell metabolism to inhibit parasite replication [27,28,29,30]. The dynamic equilibrium between Th1 and Th2 responses plays a pivotal role in modulating macrophage effector functions during *T. gondii* infection. Th1 cells activate macrophages through two principal mechanisms: (1) secretion of pro-inflammatory cytokines (IFN-γ, TNF-α, IL-2) and (2) promotion of IgG2a antibody production. Conversely, Th2 cells stimulate anti-inflammatory cytokine secretion (IL-4, IL-5, IL-6, IL-10) and IgG1 production, resulting in the suppression of macrophage functions. Notably, low levels of IL-4 and IL-10 were detected in this study, suggesting partial Th2 activation. IL-4 antagonizes IFN-γ activity in murine models, enhancing IgG1 synthesis and FcR2 expression on macrophages while suppressing IgG2a [31,32,33,34]. Additionally, IL-4 subverts host defense against *T. gondii* by inhibiting Th1 immunity—an immunosuppressive effect that may predispose one to pregnancy complications [35].

In this study, subcutaneous administration with rGRA12 resulted in oocyst shedding reductions of 20.10% in Group G3 and 27.88% in Group G4. In contrast, Garcia reported a 90.8% reduction in oocyst shedding in cats immunized intranasally with crude *T. gondii* antigens, while Zulpo observed an 86.7% reduction following intranasal administration of recombinant ROP2 (rROP2) protein. These comparisons highlight the significant impact that the route of immunization can have on the effectiveness of oocyst suppression [36,37].

As *T. gondii* oocyst formation primarily occurs in the intestinal epithelial cells of felines [38], intranasal vaccination may offer distinct advantages due to the favorable mucosal environment—featuring neutral pH, minimal enzymatic activity, and a large surface area—which helps preserve antigen stability and reduce required dosages. Additionally, intranasal immunization stimulates the common mucosal immune system, inducing dual mucosal and systemic immune responses that enhance intestinal secretory IgA (sIgA) production [39,40]. While sIgA is well-studied in humans, experimental findings also demonstrate its ability to inhibit *T. gondii* invasion of intestinal cells and limit parasite replication both in vitro and in vivo [41,42], indicating conserved protective functions across mammalian species.

The addition of adjuvants represents a critical strategy for enhancing vaccine immunogenicity while maintaining safety. ISA 201, an oil-based adjuvant from the Montanide™ ISA series, offers ready-to-use convenience, low viscosity, and excellent injectability, while demonstrating an absence of pyrogenic reactions, granuloma formation, or cyst induction [43]. Zulpo immunized cats with crude *T. gondii* rhoptry proteins adjuvanted with Quil-A and observed a reduction in oocyst shedding rates from 98.4% to 53.0%. Furthermore, immunization with the recombinant ROP2 protein plus Quil-A adjuvant resulted in an 86.7% decrease in oocyst shedding [37,44]. These findings suggest that, beyond antigen optimization, the strategic selection of adjuvants is critical for enhancing vaccine immunogenicity and promoting the production of immunologically active molecules, ultimately eliciting more robust and durable protective immunity.

The delayed mortality observed in cats vaccinated with rGRA12 may indicate the establishment of a regulated Th1/Th2 immune response. While Th1-related cytokines such as IFN-γ are essential for parasite control, excessive Th1 activity has been associated with early death during toxoplasmosis. In contrast, Th2 cytokines like IL-4 and IL-10 help moderate this response, reducing the risk of acute-phase mortality by dampening excessive inflammation [45,46]. This immunological balance likely facilitates effective parasite elimination while minimizing tissue damage. Consistent with this interpretation, analysis of tissue samples showed significantly lower *T. gondii* burdens in the vaccinated groups (G3 and G4) compared to controls (G1) (*p* < 0.05), confirming the systemic protective effect of rGRA12 immunization.

Studies have demonstrated that the peak period of oocyst shedding in cats infected with *T. gondii* occurs between 5 and 8 days post-infection, with oocyst shedding spanning 7 to 20 days [44,47]. Therefore, in this study, fecal oocyst shedding was monitored in all experimental cat groups for 20 days following infection with RH strain tachyzoites. After immunization with rGRA12 protein, G4 stopped shedding oocysts by day 16, showing an earlier cessation and a shorter shedding duration compared to other groups. However, as oocyst shedding persisted in all groups, potential public health risks remain. We did not examine possible differences in oocyst infectivity between treatment groups.

While the observed differences in oocyst shedding counts between experimental groups were close on a logarithmic scale, we rigorously verified our quantification through repeated measurements. After systematically excluding potential technical artifacts, we hypothesize that although the reduction in oocyst shedding post-immunization is not statistically significant on a logarithmic scale, immunization may still exert subtle effects by potentially damaging oocyst structure, attenuating infectivity, impairing sporulation efficiency, or compromising sporozoite viability.

This study demonstrated that rGRA12 protein reduced parasite burden in cats; however, it did not assess histopathological changes in feline tissues to evaluate potential mitigation of *T. gondii*-induced organ damage. The current study employed a relatively small sample size (*n* = 3), which may compromise statistical power (reducing the ability to detect true effects, resulting in unstable findings) and generalizability (limiting the coverage of diverse population characteristics and variations). Furthermore, although *T. gondii* oocyst development primarily occurs in the feline small intestine, this investigation measured only systemic IgG and its subclasses, without examining mucosal IgA responses. Moreover, the immunization strategy employed in this study was relatively restricted. In contrast to subcutaneous administration, intranasal vaccination has been shown to elicit both mucosal and systemic immune responses, significantly enhancing intestinal secretory IgA (sIgA) production, which may confer superior efficacy in reducing oocyst shedding. This study utilized only a single *T. gondii* strain genotype for immune protection analysis. Whether the rGRA12 protein can demonstrate comparable protective effects against *T. gondii* strains of different genotypes remains to be determined. 

## 5. Conclusions

Subcutaneous immunization with the rGRA12 protein formulated with ISA 201 adjuvant elicited a mixed immune response skewed toward Th1 in cats, prolonging survival and decreasing *T. gondii* loads in critical organs such as the liver, lungs, and brain. These results demonstrate the vaccine candidate’s dual protective capacity: it can effectively limit parasite replication while enhancing host immune defense. Future investigations should explore combinatorial approaches incorporating multiple *T. gondii* antigens with highly immunogenic epitopes, while also evaluating diverse adjuvant formulations and alternative immunization routes to optimize protective immunity. Additionally, future research should specifically examine morphological and infectivity changes in oocysts post-immunization, as well as investigate whether vaccination can suppress secondary oocyst shedding in cats upon re-infection. This approach may also be adapted for kittens, thereby reducing public health risks.

## Figures and Tables

**Figure 1 vaccines-13-00851-f001:**
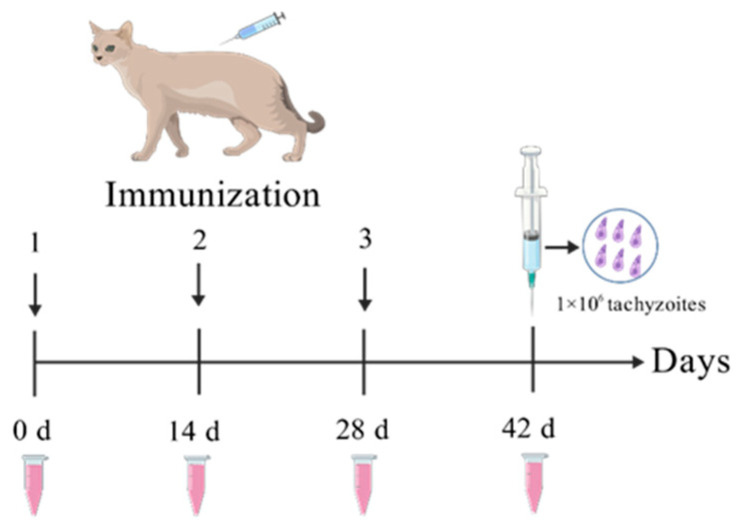
rGRA12 immunization procedure diagram.

**Figure 2 vaccines-13-00851-f002:**
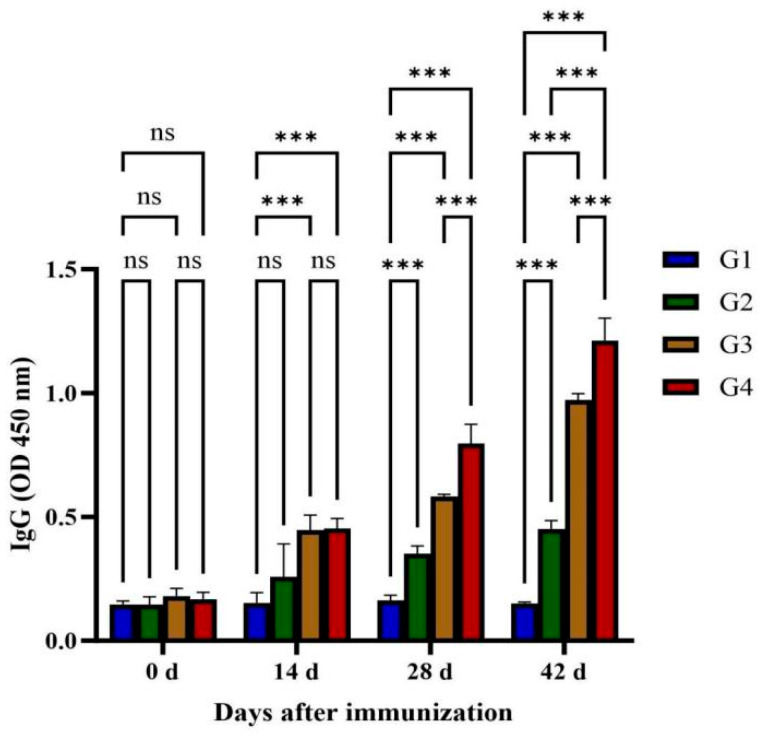
Quantitative detections of total specific anti-GRA12 IgG antibodies in cat sera. Groups were defined as follows: G1: PBS group, G2: ISA 201 group, G3: rGRA12 group, and G4: rGRA12 + ISA 201 group. ns *p* > 0.05; *** *p* < 0.001.

**Figure 3 vaccines-13-00851-f003:**
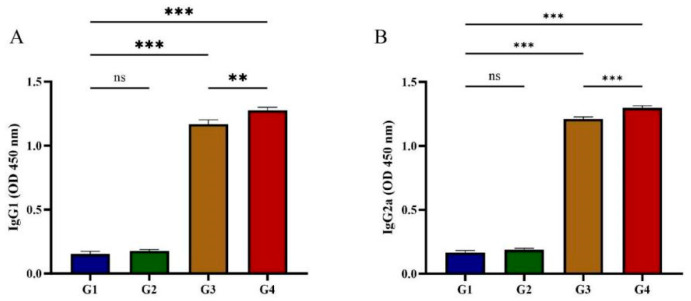
Determination of IgG isotypes in post-immunization cat serum. (**A**) Post-immunization IgG1 changes in feline serum. (**B**) Post-immunization IgG2a changes in feline serum. Groups were defined as follows: G1: PBS group, G2: ISA 201 group, G3: rGRA12 group, and G4: rGRA12 + ISA 201 group. ns *p* > 0.05; ** *p* < 0.01; *** *p* < 0.001.

**Figure 4 vaccines-13-00851-f004:**
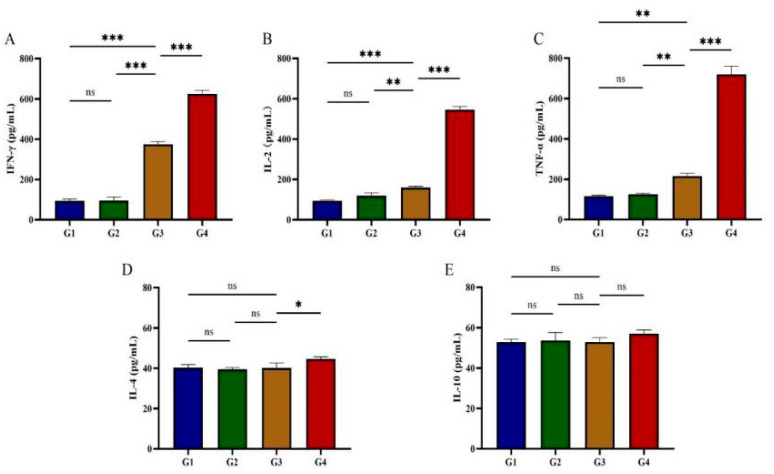
Cytokine production in post-immunization cat serum. (**A**) Post-immunization IFN-γ changes in feline serum. (**B**) Post-immunization IL-2 changes in feline serum. (**C**) Post-immunization TNF-α changes in feline serum. (**D**) Post-immunization IL-4 changes in feline serum. (**E**) Post-immunization IL-10 changes in feline serum. Groups were defined as follows: G1: PBS group, G2: ISA 201 group, G3: rGRA12 group, and G4: rGRA12 + ISA 201 group. ns *p* > 0.05; * *p* < 0.05; ** *p* < 0.01; *** *p* < 0.001.

**Figure 5 vaccines-13-00851-f005:**
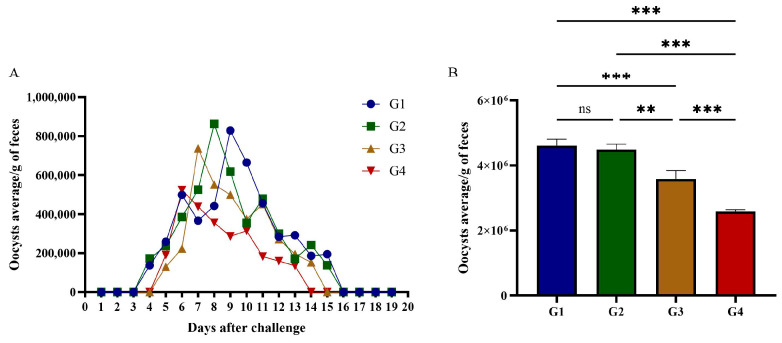
Oocyst shedding in cats challenged with tachyzoites of *T. gondii*. (**A**) Excretion of *T. gondii* oocysts in cats immunized with rGRA12 protein. (**B**) The total number of *T. gondii* oocysts excreted by each group of cats. Groups were defined as follows: G1: PBS group, G2: ISA 201 group, G3: rGRA12 group, and G4: rGRA12 + ISA 201 group. ns *p* > 0.05; ** *p*< 0.01; *** *p* < 0.001.

**Figure 6 vaccines-13-00851-f006:**
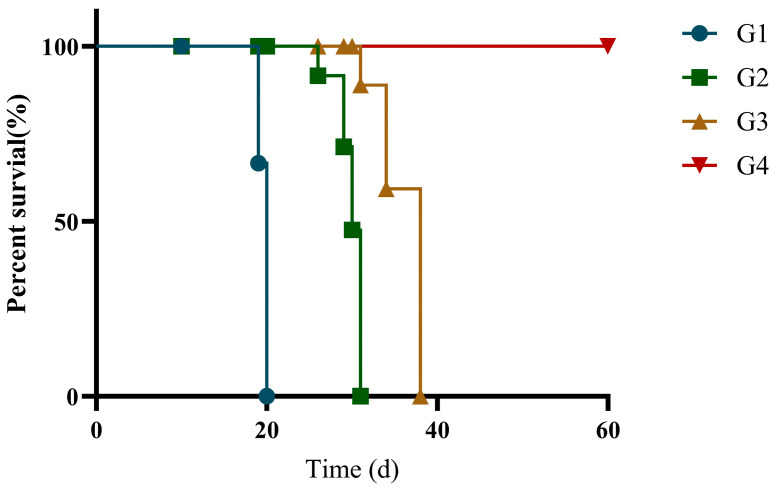
Survival rate of the immunized cats. Groups were defined as follows: G1: PBS group, G2: ISA 201 group, G3: rGRA12 group, and G4: rGRA12 + ISA 201 group. Log-rank test demonstrated an overall significant difference in survival rate between all groups of mice (χ^2^ = 68.21, df = 3, *p* < 0.0001).

**Figure 7 vaccines-13-00851-f007:**
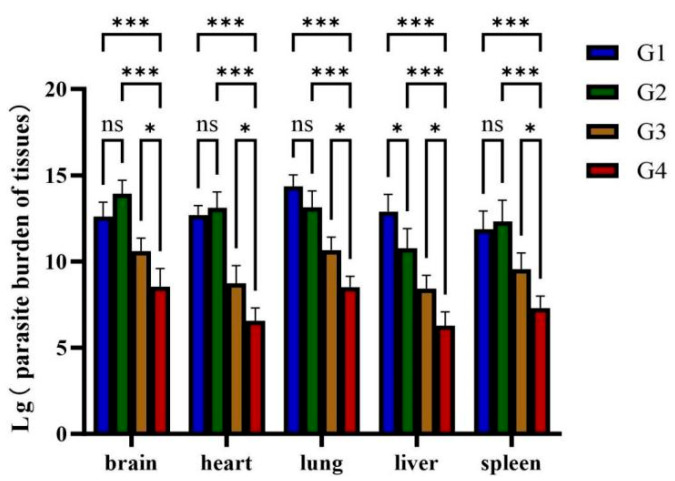
Parasite burden in various organs of cats. Groups were defined as follows: G1: PBS group, G2: ISA 201 group, G3: rGRA12 group, and G4: rGRA12 + ISA 201 group. ns *p* > 0.05; * *p* < 0.05; *** *p* < 0.001.

**Table 1 vaccines-13-00851-t001:** Immunization and challenge of cats.

Group	Immune Pathway	Immune Dose	Vaccination Schedule (Weeks)	Number of Immunizations	*T. gondii* Strain
G1	Subcutaneous injection	1 mL	2	3	RH
G2		1 mL			
G3		1 mL (200 μg)			
G4		1 mL (200 μg)			

*T. gondii* tachyzoites were inoculated intraperitoneally (IP) into experimental animals at a dose of 1 × 10^6^.

**Table 2 vaccines-13-00851-t002:** Assessment of oocyst shedding in post-immunization cats.

Group	Incubation Period (d)	Oocyst Shedding Period (d)	Total Number of Oocyst Shedding (Mean ± SE)	Inhibition Rate (%)
G1	3	12	4,607,685 ± 200,786.67	/
G2	3	12	4,486,855 ± 167,599.45	2.62
G3	4	10	3,584,980 ± 256,171.38	20.10
G4	4	9	2,585,635 ± 56,333.95	27.88

## Data Availability

The datasets supporting the findings of this study are available within the article.

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
