# Peer review of "Immunogenicity and Protective Efficacy of a Recombinant *Toxoplasma gondii* GRA12 Vaccine in Domestic Cats"

_vaccines, 2025, doi:10.3390/vaccines13080851_

Round 1

Reviewer 1 Report

Comments and Suggestions for Authors

The article describes a well-designed experimental study evaluating the immunogenicity and protective efficacy of a recombinant T. gondii GRA12 protein vaccine in domestic cats. The authors present a detailed methodology encompassing immunization, challenge, immune response profiling (antibody titers, cytokines), oocyst excretion, and parasite burden in tissues. The results demonstrate that rGRA12, especially when combined with the ISA 201 adjuvant, triggers a mixed but Th1-skewed immune response, significantly reduces oocyst shedding, lowers tissue parasite load, and prolongs survival in challenged animals. The findings contribute to ongoing efforts to develop effective vaccines against feline toxoplasmosis, which has public health and veterinary relevance.

In my opinion, the course of the experiment is relatively well-designed; however, the manuscript would benefit from improvements in language editing, formatting, and a more in-depth discussion, as these aspects currently limit the overall clarity and impact of the study. Therefore, I have listed my specific comments and suggestions in the points below.

  1. There are several instances of grammatical and stylistic issues throughout the manuscript. For example, phrases like "dual protective capacityeffectively limit" (line 329) appear to be editing artifacts. A thorough proofreading is needed to correct typographical errors and enhance clarity!!!
    Improve sentence flow in the Discussion section, where some sentences are long and difficult to follow.
  2. The units (e.g., µg, copies/µL) and terms (e.g., “tachyzoites”, “tachyzoite”) should be standardized throughout the manuscript. Use consistent abbreviations and define them upon first use (e.g., "OPG" should be defined where first introduced, not just in the methods).
  3. Figures referenced (e.g., Fig 1–6) are described in text, but images were not visible in the reviewed version. Ensure high-quality, clearly labeled figures and legends are included in the final submission. Table 1 formatting is misaligned (lines 116–117). Reformat to ensure columns are readable and all data are correctly positioned.
  4. Although the statistical approach is briefly described (ANOVA, Tukey, etc.), some results lack precise statistical annotations in the figures (e.g., exact p-values or post hoc results). Please, consider including these in figure legends or supplemental material.
  5. In my opinion the discussion does a good job example interpreting the results; however, the limitations of the study should be acknowledged. For instance:

-The relatively small group sizes (n=3) may limit the statistical power and generalizability.

-Only the RH strain of T. gondii was used – an extremely virulent strain. It would be helpful to note that further studies should include lower-virulence strains to better reflect field conditions.

-The route of administration (subcutaneous) is discussed in relation to mucosal immunity, but no mucosal immune parameters (e.g., sIgA) were measured – this should be mentioned as a gap or future direction.

  1. The last sentence includes a typographical issue: “dual protective capacityeffectively limit parasite replication…” → should be corrected for readability!!!

The conclusion could better emphasize the translational potential or next steps (e.g., field trials, formulation optimization, testing in younger kittens, etc.).

Author Response

Comments 1:There are several instances of grammatical and stylistic issues throughout the manuscript. For example, phrases like "dual protective capacityeffectively limit" (line 329) appear to be editing artifacts. A thorough proofreading is needed to correct typographical errors and enhance clarity!!! Improve sentence flow in the Discussion section, where some sentences are long and difficult to follow.

These results demonstrate the vaccine candidate's dual protective capacity, it can effectively limit parasite replication while enhancing host immune defense.

The dynamic equilibrium between Th1 and Th2 responses plays a pivotal role in modulating macrophage effector functions during T. gondii infection. Th1 cells activate macrophages through two principal mechanisms: (1) secretion of pro-inflammatory cytokines (IFN-γ, TNF-α, IL-2) and (2) promotion of IgG2a antibody production. Conversely, Th2 cells stimulate anti-inflammatory cytokine secretion (IL-4, IL-5, IL-6, IL-10) and IgG1 production, resulting in suppression of macrophage functions. Notably, low levels of IL-4 and IL-10 were detected in this study, suggesting partial Th2 activation.

Additionally, IL-4 subverts host defense against T. gondii by inhibiting Th1 immunity-an immunosuppressive effect that may predispose to pregnancy complications [35]. 

Response: We thank the reviewer #1 who gave pertinent suggestions. Therefore, during our discussion, we have revised these sentences to facilitate better readability for readers. Mention exactly where in the revised manuscript this change can be found in line 457-458, 371-378 and 380-382.

Comments 2:The units (e.g., µg, copies/µL) and terms (e.g., “tachyzoites”, “tachyzoite”) should be standardized throughout the manuscript. Use consistent abbreviations and define them upon first use (e.g., "OPG" should be defined where first introduced, not just in the methods).

Response: Thank you for pointing this out. Therefore, we have standardized the format for unit usage and removed the abbreviation "OPG" while retaining its definition. Thank you for your understanding. Mention exactly where in the revised manuscript this change can be found in line 82 and 173-181.

Comments 3Figures referenced (e.g., Fig 1–6) are described in text, but images were not visible in the reviewed version. Ensure high-quality, clearly labeled figures and legends are included in the final submission. Table 1 formatting is misaligned (lines 116–117). Reformat to ensure columns are readable and all data are correctly positioned.

Response: Thank you for pointing this out. Therefore, we ensure that all content to be finally submitted consists of high-quality graphics and legends with clear labels, and have aligned the format of the content in lines 166-167.

Comments 4Although the statistical approach is briefly described (ANOVA, Tukey, etc.), some results lack precise statistical annotations in the figures (e.g., exact p-values or post hoc results). Please, consider including these in figure legends or supplemental material.

Response: Thank you for pointing this out. Therefore, we have added precise statistical annotations to the figure legends. Mention exactly where in the revised manuscript this change can be found in line 327-331.

Comments 5In my opinion the discussion does a good job example interpreting the results; however, the limitations of the study should be acknowledged. For instance:-The relatively small group sizes (n=3) may limit the statistical power and generalizability.

-Only the RH strain of T. gondii was used , -an extremely virulent strain. It would be helpful to note that further studies should include lower-virulence strains to better reflect field conditions.

-The route of administration (subcutaneous) is discussed in relation to mucosal immunity, but no mucosal immune parameters (e.g., sIgA) were measured , -this should be mentioned as a gap or future direction.

This study demonstrated that the rGRA12 protein reduced parasite burden in cats; however, it did not assess histopathological changes in feline tissues to evaluate potential mitigation of T. gondii induced organ damage. The current study employed a relatively small sample size (n=3), which may compromise statistical power (reducing the ability to detect true effects and resulting in unstable findings) and generalizability (limiting the coverage of diverse population characteristics and variations). Furthermore, while T. gondii oocyst development primarily occurs in the feline small intestine, the investigation only measured systemic IgG and its subclasses, without examining mucosal IgA responses. Moreover, the immunization strategy employed in this study was relatively restricted. In contrast to subcutaneous administration, intranasal vaccination has been shown to elicit both mucosal and systemic immune responses, significantly enhancing intestinal secretory IgA (sIgA) production, which may confer superior efficacy in reducing oocyst shedding. This study utilized only a single T. gondii strain genotype for immune protection analysis. Would the rGRA12 protein demonstrate comparable protective effects against T. gondii strains of different genotypes?

Response: Thank you for pointing this out. Therefore, during our discussion, we have added the section on the limitations of this article. Mention exactly where in the revised manuscript this change can be found in line 438-452.

Comments 6The last sentence includes a typographical issue: “dual protective capacityeffectively limit parasite replication…” → should be corrected for readability!!!

The conclusion could better emphasize the translational potential or next steps (e.g., field trials, formulation optimization, testing in younger kittens, etc.).

These results demonstrate the vaccine candidate's dual protective capacity, it can effectively limit parasite replication while enhancing host immune defense.

Future investigations should explore combinatorial approaches incorporating multiple T. gondii antigens with highly immunogenic epitopes, while evaluating diverse adjuvant formulations and alternative immunization routes to optimize protective immunity. Additionally, future research will specifically examine morphological and infectivity changes in oocysts post-immunization, as well as investigate whether vaccination can suppress secondary oocyst shedding in cats upon re-infection. This approach aims to adapt the strategy for kittens, thereby reducing public health risks.

Response: Thank you for pointing this out. Therefore, we have revised this sentence and added prospects for the future in the conclusion. Mention exactly where in the revised manuscript this change can be found in line 457-458 and 458-465.

Reviewer 2 Report

Comments and Suggestions for Authors

The manuscript, “Immunogenicity and Protective Efficacy of a Recombinant Toxoplasma gondii GRA12 Vaccine in Domestic Cats” by Yang et al., describes the evaluation of vaccine efficacy against T. gondii. The authors used GRA12 protein of T. gondii as immunogen and subcutaneously immunized various groups of cats for 42 days. The analysis of specific antibodies (IgG, IgG1, and IgG2a) and cytokines (IFN-γ, IL-2, TNF-α, IL-4, IL-10) indicated elevated levels of IgG, IFN-γ, IL-2, TNF-α in cats immunized with GRA12 protein-adjuvent formulation. Additionally, post immunization infection with T. gondii resulted in 100% survival and reduced parasite burden in the tissues from various organs. The manuscript is written well and presents all the results clearly. The authors are suggested to address the following concerns:

1. Although authors claim to have achieved 100% survival rate, the difference between the measured oocyst shedding counts across the various cat groups (G1, G2, G3, and G4) post gondiiinfection  remains little insignificant. In fact, from 4.61×106 (G1) to 2.59×106 (G4) they all are very close on a log scale. Authors should cross check their oocyte quantification and explain it in the discussion part of the manuscript.

2. The authors used GRA12 fusion protein as immunogen to vaccinate the cats. However, there is no detail about the protein sequence (at least accession No.) or biochemical details (gel picture showing protein size and purity) of the protein in the manuscript. Please include them appropriately (methods and supplementary result) in the manuscript.

Author Response

Comments 1:Although authors claim to have achieved 100% survival rate, the difference between the measured oocyst shedding counts across the various cat groups (G1, G2, G3, and G4) post gondiiinfection  remains little insignificant. In fact, from 4.61×106 (G1) to 2.59×106 (G4) they all are very close on a log scale. Authors should cross check their oocyte quantification and explain it in the discussion part of the manuscript.

While the observed differences in oocyst shedding counts between experimental groups showed close proximity on a logarithmic scale, we rigorously verified our quantification through repeated measurements. After systematically excluding potential technical artifacts, we hypothesize that although the reduction in oocyst shedding post-immunization is not statistically significant on a logarithmic scale, the immunization may still exert subtle effects by potentially damaging oocyst structure, attenuating infectivity, impairing sporulation efficiency, or compromising sporozoite viability.

Response: We thank the reviewer #2 who gave pertinent suggestions. Therefore, during our discussion, we have increased the likelihood of this result occurring, providing a direction for the subsequent experiments. Mention exactly where in the revised manuscript this change can be found in line 431-437.

Comments 2:The authors used GRA12 fusion protein as immunogen to vaccinate the cats. However, there is no detail about the protein sequence (at least accession No.) or biochemical details (gel picture showing protein size and purity) of the protein in the manuscript. Please include them appropriately (methods and supplementary result) in the manuscript.

2.4. Construction of plasmids, expression and purification of rGRA12

The protocol for obtaining rGRA12 was conducted following the method described by Wang[20]. The DNA sequence of the T. gondii GRA12 gene was obtained from the GenBank database (accession number: FJ011096.1). Genomic DNA was isolated from T.gondii RH strain tachyzoites and used as a template to amplify the GRA12 gene via standard PCR. Based on its sequence, the predicted molecular weight of GRA12 antigen is 47.8 kDa.The 1311 bp GRA12 fragment was amplified using primers GRA12F (5'-TGAGCTCATCATGAGGGCGATCGTGGCATCGACG-3', SacI site underlined) and GRA12R (5'-CAAGCTTGTTGTGTTTGCTGCCTGCAGAGCCGCG-3', HindIII site underlined), then cloned into the pET-28a vector via SacI/HindIII restriction sites. The recombinant plasmids were verified by SacI/HindIII double digestion and sequencing of the gel-purified fragments. Sequence-confirmed positive clones were subsequently transformed into E. coli BL21 for protein expression. E. coli BL21(DE3) harboring pET-28a-GRA12 was inoculated from a 5 ml starter culture into 500 ml of kanamycin-supplemented LB medium. The culture was grown at 37°C with 170 rpm shaking to OD600 0.4-0.5, at which Comments protein expression was induced with 0.1% IPTG. Following 4 hours of continued shaking under identical conditions, the bacterial cells were harvested for subsequent protein purification. The purified recombinant GRA12 protein (47.8 kDa) was subsequently used for feline immunization.

Reference:

[20] Wang YH. Prokaryotic expression and immunogenicity study of Toxoplasma gondii dense granule protein GRA12 . 2017.

Response: Thank you for pointing this out. Therefore, we have supplemented the expression method of this protein in the manuscript. Since this part of the content is derived from our team's previous research work and the focus of this study is to evaluate the immune protective effect of the protein in cats rather than on protein expression-related content, we have not further elaborated on the presentation of relevant results. Mention exactly where in the revised manuscript this change can be found in line 132-150 and 526-527.

Reviewer 3 Report

Comments and Suggestions for Authors

Please address the following comments: This work can be accepted after minor revision.

  1. Line 80: A detailed objective of the study might enhance the readability
  2. Line 90: discuss the rationale behind the selection of sample size
  3. Line 107: please include any reference orelse provide the detailed protocol in suplementory document
  4. Please discuss more about the use of adjuvants, including existing literature, rationale, etc
  5. Include the limitations of the study
  6. A detailed future perspective is required in the conclusion section
  7. The author can provide a schematic representation of the dosing regimen

Author Response

Comments 1:Line 80: A detailed objective of the study might enhance the readability

Overall, this study aims to demonstrates the immunogenicity and protective potential of the recombinant GRA12 (rGRA12) subunit vaccine including its capacity to reduce T. gondii transmission in felids and mitigate host damage, providing a scientific basis for further Toxoplasma vaccine development.

Response: We thank the reviewer #3 for this constructive comment and suggestion. Therefore, we have added detailed research objectives. Mention exactly where in the revised manuscript this change can be found in line 96-100.

Comments 2:Line 90: discuss the rationale behind the selection of sample size

To maximize animal welfare while maintaining scientific validity, we implemented the 3Rs principle (Replacement, Reduction, and Refinement) throughout our experimental design. This included using the minimum number of animals required for statistically significant results,  optimizing protocols to minimize discomfort.

Response: Thank you for pointing this out. Therefore, we have added the principle of selecting the sample size. Mention exactly where in the revised manuscript this change can be found in line 110-113.

Comments 3:Line 107: please include any reference orelse provide the detailed protocol in suplementory document

Reference:

[19] Liang XH. Biological roles of de novo synthesis of saturated fatty acids and phosphatidylcholine in T. gondii. Huazhong Agricultural University, 2022.

Response: Thank you for pointing this out. Therefore, we have added more reference materials. Mention exactly where in the revised manuscript this change can be found in line 131 and 524-525.

Comments 4:Please discuss more about the use of adjuvants, including existing literature, rationale, etc

The addition of adjuvants represents a critical strategy for enhancing vaccine immunogenicity while maintaining safety. ISA 201, an oil-based adjuvant from the Montanide™ ISA series, exhibits ready-to-use convenience, low viscosity, and excellent injectability, while demonstrating an absence of pyrogenic reactions, granuloma formation, or cyst induction[43]. Zulpo immunized cats with crude T. gondii rhoptry proteins adjuvanted with Quil-A, observing a reduction in oocyst shedding rates from 98.4% to 53.0%. Furthermore, immunization with the recombinant ROP2 protein plus Quil-A adjuvant resulted in an 86.7% decrease in oocyst shedding[44, 45]. These findings suggest that, beyond antigen optimization, the strategic selection of adjuvants is critical for enhancing vaccine immunogenicity and promoting the production of immunologically active molecules, ultimately eliciting more robust and durable protective immunity.

Response: Thank you for pointing this out. Therefore, during our discussion, we incorporated the existing literature and principles regarding the use of adjuvants. Mention exactly where in the revised manuscript this change can be found in line 400-410.

Comments 5:Include the limitations of the study

This study demonstrated that the rGRA12 protein reduced parasite burden in cats; however, it did not assess histopathological changes in feline tissues to evaluate potential mitigation of T. gondii induced organ damage. The current study employed a relatively small sample size (n=3), which may compromise statistical power (reducing the ability to detect true effects and resulting in unstable findings) and generalizability (limiting the coverage of diverse population characteristics and variations). Furthermore, while T. gondii oocyst development primarily occurs in the feline small intestine, the investigation only measured systemic IgG and its subclasses, without examining mucosal IgA responses. Moreover, the immunization strategy employed in this study was relatively restricted. In contrast to subcutaneous administration, intranasal vaccination has been shown to elicit both mucosal and systemic immune responses, significantly enhancing intestinal secretory IgA (sIgA) production, which may confer superior efficacy in reducing oocyst shedding. This study utilized only a single T. gondii strain genotype for immune protection analysis. Would the rGRA12 protein demonstrate comparable protective effects against T. gondii strains of different genotypes?

Response: Thank you for pointing this out. Therefore, during our discussion, we identified the limitations of this article and enhanced its readability. Mention exactly where in the revised manuscript this change can be found in line 438-452.

Comments 6:A detailed future perspective is required in the conclusion section

 Future investigations should explore combinatorial approaches incorporating multiple T. gondii antigens with highly immunogenic epitopes, while evaluating diverse adjuvant formulations and alternative immunization routes to optimize protective immunity. Additionally, future research will specifically examine morphological and infectivity changes in oocysts post-immunization, as well as investigate whether vaccination can suppress secondary oocyst shedding in cats upon re-infection. This approach aims to adapt the strategy for kittens, thereby reducing public health risks.

Response: Thank you for pointing this out. Therefore, during our discussion, we incorporated the future outlook presented in this article, aiming to enhance the immune efficacy of the vaccine. Mention exactly where in the revised manuscript this change can be found in line 458-465.

Comments 7:The author can provide a schematic representation of the dosing regimen.

Figure 1. rGRA12 immunization procedure diagram

Response: Thank you for pointing this out. We agree with this comment. Therefore, we have added the immunization flowchart. Mention exactly where in the revised manuscript this change can be found in line 163-165.

Reviewer 4 Report

Comments and Suggestions for Authors

This manuscript is written according to the Guidelines for authors of the journal, but I have the following notes:

Twelve 4-month-old British Shorthair cats or A total of twelve 6-month-old female domestic cats – which is correct, 4-month-old or 6-month-old kittens? If kittens were acclimated for 2 months before beginning the study, please clarify that in the methods.

Please describe how the group size was determined

In Table 3, the results are presented as means, but without Standard errors (SE). I recommend that the table be revised to present the results as mean ± SE.

The authors report that of all the kittens, only those from group G4 survived. However, tachyzoites were found in the organs of the surviving animals. This means that surviving kittens can still pose a danger to other cats and to the people who care for them. Has it been studied whether oocysts are found in the feces of surviving kittens? Please discuss the limitations of the study if examination of viable oocysts in feces was not performed.

In section “Conclusion” the word “capacityeffectively” should be “capacity effectively.”

Author Response

Comments 1:Twelve 4-month-old British Shorthair cats or A total of twelve 6-month-old female domestic cats – which is correct, 4-month-old or 6-month-old kittens? If kittens were acclimated for 2 months before beginning the study, please clarify that in the methods.

The animals were previously monitored for 2 months prior to the beginning of the experiment.

Response: We are very grateful to reviewer #4 who carefully gave pertinent and constructive suggestions for us to improve the MS. Therefore, we have revised the inappropriate descriptions to ensure the completeness of the article. Mention exactly where in the revised manuscript this change can be found in line 116-117.

Comments 2:In Table 3, the results are presented as means, but without Standard errors (SE). I recommend that the table be revised to present the results as mean ± SE.

Response: Thank you for pointng this out. Therefore, in Table 1, we have added the Standard Errors (SE). Mention exactly where in the revised manuscript this change can be found in line 314-315. 

Comments 3:The authors report that of all the kittens, only those from group G4 survived. However, tachyzoites were found in the organs of the surviving animals. This means that surviving kittens can still pose a danger to other cats and to the people who care for them. Has it been studied whether oocysts are found in the feces of surviving kittens? Please discuss the limitations of the study if examination of viable oocysts in feces was not performed.

Studies have demonstrated that the peak period of oocyst shedding in cats infected with T. gondii occurs between 5 to 8 days post-infection, with the oocyst spanning 7 to 20 days [48, 44]. Therefore, in this study, fecal oocyst shedding was monitored in all experimental cat groups for 20 days following infection with RH strain tachyzoites. After immunization with rGRA12 protein, Group G4 stopped shedding oocysts by day 16, showing an earlier cessation and shorter shedding duration compared to other groups. However, oocyst shedding persisted in all groups, potential public health risks remain. We did not examine possible differences in oocyst infectivity between treatment groups.

Response: Thank you for pointing this out. Therefore, we explained the cycle of cat egg oocysts discharge, and also described the infection risks for the environment and the surrounding population. Mention exactly where in the revised manuscript this change can be found in line 423-430.

Comments 4:In section “Conclusion” the word “capacityeffectively” should be “capacity effectively.”

Response: Thank you for pointing this out. Therefore, we have amended the error. Mention exactly where in the revised manuscript this change can be found in line 457.

Round 2

Reviewer 4 Report

Comments and Suggestions for Authors

I am satisfied with the corrections made in the revised text.